# COVID-19 Pathology in the Lung, Kidney, Heart and Brain: The Different Roles of T-Cells, Macrophages, and Microthrombosis

**DOI:** 10.3390/cells11193124

**Published:** 2022-10-04

**Authors:** Tino Emanuele Poloni, Matteo Moretti, Valentina Medici, Elvira Turturici, Giacomo Belli, Elena Cavriani, Silvia Damiana Visonà, Michele Rossi, Valentina Fantini, Riccardo Rocco Ferrari, Arenn Faye Carlos, Stella Gagliardi, Livio Tronconi, Antonio Guaita, Mauro Ceroni

**Affiliations:** 1Department of Neurology and Neuropathology, Golgi-Cenci Foundation, Abbiategrasso, 20081 Milan, Italy; 2Department of Rehabilitation, ASP Golgi-Redaelli, Abbiategrasso, 20081 Milan, Italy; 3Department of Public Health, Experimental and Forensic Medicine, University of Pavia, 27100 Pavia, Italy; 4Unit of Biostatistics, Golgi-Cenci Foundation, Abbiategrasso, 20081 Milan, Italy; 5Laboratory of Neurobiology and Neurogenetic, Golgi-Cenci Foundation, Abbiategrasso, 20081 Milan, Italy; 6Unit of Molecular Biology and Transcriptomics IRCCS Mondino Foundation, 27100 Pavia, Italy; 7Department of Forensic Medicine, IRCCS Mondino Foundation, 27100 Pavia, Italy

**Keywords:** COVID-19, SARS-CoV-2, lung, kidney, heart, brain, inflammation, elderly, neuropathology

## Abstract

Here, we aim to describe COVID-19 pathology across different tissues to clarify the disease’s pathophysiology. Lungs, kidneys, hearts, and brains from nine COVID-19 autopsies were compared by using antibodies against SARS-CoV-2, macrophages-microglia, T-lymphocytes, B-lymphocytes, and activated platelets. Alzheimer’s Disease pathology was also assessed. PCR techniques were used to verify the presence of viral RNA. COVID-19 cases had a short clinical course (0–32 days) and their mean age was 77.4 y/o. Hypoxic changes and inflammatory infiltrates were present across all tissues. The lymphocytic component in the lungs and kidneys was predominant over that of other tissues (*p* < 0.001), with a significantly greater presence of T-lymphocytes in the lungs (*p* = 0.020), which showed the greatest presence of viral antigens. The heart showed scant SARS-CoV-2 traces in the endothelium–endocardium, foci of activated macrophages, and rare lymphocytes. The brain showed scarce SARS-CoV-2 traces, prominent microglial activation, and rare lymphocytes. The pons exhibited the highest microglial activation (*p* = 0.017). Microthrombosis was significantly higher in COVID-19 lungs (*p* = 0.023) compared with controls. The most characteristic pathological features of COVID-19 were an abundance of T-lymphocytes and microthrombosis in the lung and relevant microglial hyperactivation in the brainstem. This study suggests that the long-term sequelae of COVID-19 derive from persistent inflammation, rather than persistent viral replication.

## 1. Introduction

Coronaviruses (CoVs) are enveloped, positive-sense, single-stranded RNA viruses. They are characterized by frequent genomic recombination, high prevalence, and wide distribution, and they typically infect the respiratory and digestive tracts of several animal species, as well as humans. In certain geographical areas, the close interface between humans and animals facilitates zoonotic transmissions, resulting in the emergence of new human CoVs [1]. Common human CoVs include two alpha-CoVs (HCoV-229E and HCoV-NL63) and two beta-CoVs (HCoV-OC43 and HCoV-HKU1) which cause mild, self-limiting upper-respiratory-tract infections [2]. Contrarily, SARS-CoV and MERS-CoV are two highly pathogenic beta-CoVs, which have been identified as the etiologic agents of SARS (severe acute respiratory syndrome) and MERS (middle east respiratory syndrome) outbreaks in 2002 and 2012, respectively. In autumn 2019, an outbreak of severe pneumonia of unidentified etiology was reported in Wuhan, Hubei Province, China. On 9 January 2020, China announced the identification of a novel beta-CoV as the etiologic agent of the outbreak. The virus was subsequently named SARS-CoV-2 by the International Committee on Taxonomy of Viruses. It causes a systemic disease identified as coronavirus 2019 disease (COVID-19). The mortality rates due to SARS-CoV (9.6%) and MERS-CoV (34.3%) are significantly higher than that of SARS-CoV-2 infection (4.4%), but the latter is more easily transmittable, which explains its rapid spread and the massive number of cases worldwide [3]. SARS-CoV-2 represents a “perfect pandemic virus”, surpassing SARS-CoV and MERS-CoV in terms of infected people, geographical expansion, and deaths [4]. On 1 March 2020, COVID-19 was declared a pandemic by the World Health Organization (WHO) and remains to this day a global threat to public health (WHO-March 2020). By August 2022, there had been 589,680,368 confirmed cases of COVID-19, with 6,436,519 deaths reported to the WHO (https://covid19.who.int/ accessed on 20 August 2022). Although the respiratory system is undoubtedly the main target of SARS-CoV-2, the infection is characterized by a broad spectrum of clinical manifestations denoting a multiple-organ disease, which has been confirmed by pathologic studies [5,6,7,8]. The involvement of different organs and extent of the lesions can be considered reliable prognostic factors for adverse outcomes in COVID-19 patients and for the development of post-acute COVID-19 syndrome [9]. Indeed, post-acute COVID-19 syndrome, divided into subacute or ongoing symptomatic COVID-19 (4–12 weeks after disease onset) and chronic or post-COVID syndrome (beyond 12 weeks), may affect the lungs, kidneys, heart, and brain with variable severity [9]. Pulmonary outcomes range from a chronic cough to respiratory insufficiency due to fibrosis [10], renal dysfunction may persist and lead to chronic kidney disease [11], heart complications include arrhythmias and heart failure [10,12], and the so-called “brain fog” affects cognitive functions after the acute phase of the disease [13].

Damage to the lungs is typically marked by diffuse alveolar damage (DAD) of variable degrees and stages (acute–proliferative–fibrotic), with edema and hyaline membrane formation, cytopathic features, and hyperplasia of type-2 pneumocytes, and the presence of SARS-CoV-2 antigens, accompanied by macrophage and lymphocyte infiltration, organizing pneumonia, and frequent superimposed bacterial infection. Moreover, vascular damage is often described with the formation of microthrombi and thrombi, hemorrhagic infarctions, and pulmonary thrombo-embolism [3,14,15,16,17,18,19,20,21,22,23,24]. DAD is a key pathophysiological mechanism of lung damage in SARS-CoV-2 infection that is present in 88% of cases; however, it is not pathognomonic for COVID-19. Indeed, DAD with prominent hyaline membrane formation is also very frequent in SARS-CoV (98%) and influenza A/H1N1 (90% in the 2009 pandemic). On the other hand, micro-thrombotic disease has been reported with a similarly high prevalence among COVID-19 (57%) and SARS (58%) cases, while it is lower for H1N1 (24%) flu cases [25].

At the kidney level, the main histological findings are acute tubular injury, inflammatory infiltrates, and microvascular occlusion of glomerular and peritubular capillaries, frequently accompanied by arteriolosclerosis and glomerular degeneration (pre-existing chronic renal disease). Transmission electron microscopy has revealed the presence of viral particles in the tubular epithelium and podocytes [8,22,23,26,27].

Regarding the heart, the most common findings are pericarditis and myocarditis with inflammatory foci associated with myocyte injury and fibrosis that may also reflect a pre-existing disease [15,16,21,23,28,29,30]. In the heart, there are only scarce molecular traces of SARS-CoV-2 [31], while macrophage infiltration is predominant with a low number of lymphocytes [32]. Other authors have reported a predominance of thrombosis and micro-thrombosis leading to ischemic injury [30,33].

From a neuropathologic standpoint, a wide range of changes are observed. In almost all postmortem evaluations, brain congestion, edema, and neuronal loss caused by severe hypoxic phenomena due to pulmonary and heart complications have been observed. Moreover, inflammatory processes are frequently described, including acute disseminated encephalomyelitis (ADEM)-like features and different patterns of immune-induced meningoencephalitis with meningeal, perivascular, or parenchymal lympho-monocytic infiltrates, while the presence of SARS-CoV-2 in the brain remains controversial. Microglial activation with microglial nodules is often detected. In this regard, it should be considered that the elderly population is the most affected by the severe form of COVID-19, and many patients had pre-existing neurocognitive disorders; thus, brain inflammation changes and consequent neurological manifestations may be greatly influenced by the presence of microglial “priming” due to neurodegeneration [34,35,36,37,38,39,40,41,42,43,44]. Vascular injuries of either the ischemic or hemorrhagic type are also reported, including macroscopic and microscopic lesions caused by clotting alterations and/or endotheliitis [35,37,40,41,42,45,46,47].

Although the liver is one of the most important immunological organs in the body and alterations in liver parameters are frequently reported in COVID-19, especially in severe cases, the pathological findings are non-specific and the impairment of liver function does not appear clinically relevant in SARS-CoV-2 infection [48,49,50].

Many assume the presence of active viral replication, not only in the lungs, but also in other organs [51,52,53], probably depending on the differential expression of angiotensin-converting enzyme 2 (ACE-2) receptors and TMPRSS-2 transmembrane protease, which are the main cellular factors involved in viral entry [54]. COVID-19 induces multi-organ damage, the pathological aspects of which are essential for understanding the pathophysiology of the acute disease, as well as its long-term manifestations. Nonetheless, comparative studies between the various organs involved are still lacking. The aims of this work are: (1) to describe how the above-mentioned organs are involved and how SARS-CoV-2 spreads and persists throughout the organism; (2) to compare the inflammatory infiltrates of the lungs, the organ massively affected by the viral invasion, with those of the kidneys, heart, and brain, which are non-primary targets for the virus; (3) to emphasize the pathological features specific for SARS-CoV-2 infection through a comparison between COVID-19 and non-COVID lungs, and between COVID-19 brains with and without neurodegenerative burden (i.e., Alzheimer’s disease—AD pathology) and non-COVID brains with and without AD pathology; and (4) to investigate the role of microthrombosis.

## 2. Materials & Methods

### 2.1. Study Design, Setting, Participants, and Clinical Data 

This is an observational study based on a cross-sectional analysis of clinical and pathological data from COVID-19 cases. The study comprises patients from elderly care units who died during the first tumultuous pandemic peak. Most of them were not hospitalized, and the availability of blood tests is scant; thus, the information obtained through a retrospective review of medical records is limited to clinical data. COVID-19 cases were subjected to forensic autopsies, ordered by the Prosecutor. Human autopsy samples were harvested and provided by the Unit of Legal Medicine and Forensic Sciences (Department of Public Health, Experimental and Forensic Medicine, University of Pavia, Pavia, Italy). All consecutive COVID-19 autopsies performed between 17 April and 4 June 2020 were considered for this study. The inclusion criteria were: (1) SARS-CoV-2 infection (Delta-variant) confirmed by a positive pharyngeal swab and (2) continuous refrigeration of the cadaver at 4 °C leading to the time of autopsy with adequate tissue preservation for histological multi-organ comparison, including the brain. Of the 15 COVID-19 autopsies performed, 9 were selected, while 6 were excluded for inadequate preservation of the brain tissue. Owing to the presence of cognitive disturbances in over half of the cases, and the fact that these disturbances worsened during COVID-19, we chose to also evaluate the presence of AD neuropathology. In addition, 10 matched controls were studied: for lung comparison, 5 cases with non-COVID pneumonia were selected from the Unit of Legal Medicine, while, for neuropathological comparison, 5 non-COVID brains were selected from the Abbiategrasso Brain Bank (ABB) at the Golgi-Cenci Foundation (Abbiategrasso, Milan, Italy), including 3 cases with AD pathology and 2 with no AD pathology. A retrospective review of medical charts was performed by two forensic medical doctors, a geriatrician, and a neurologist in order to ascertain the clinical history of the selected cases. The patients were clinically defined for the presence or absence of comorbidities, dementia, delirium, and sepsis. The DSM-5 criteria were used to define the mental state and identify any pre-existing cognitive dysfunction, namely major neurocognitive disorder (major-NCD) to indicate dementia and mild-NCD to designate mild cognitive impairment (MCI). Sepsis was considered a severe bacterial superinfection with at least one positive blood culture.

### 2.2. Autopsies and Sampling of the Organs

All autopsies were conducted respecting all the recommendations for forensic autopsy in SARS-CoV-2 infected cadavers [55]. All tissue samples were harvested and processed as previously described [34,56]. Briefly, the sampling protocol for pathological examination included 1 section per each pulmonary lobe; 5 heart sections from a mid-horizontal slice (anterior and posterior right ventricle, septum, left ventricle, and 1 epicardial coronary); and 1 section from each kidney, including the cortex and medulla (The Royal College of Pathologists 2020) [57]. For the neuropathological characterization, a total of 7 sections were considered: frontal, temporal, parietal, and occipital lobes; hippocampus–entorhinal cortex; pons; and the cerebellum. Before fixation, a small portion from the fronto-basal region was frozen for quantitative Reverse-Transcription–PCR (qRT-PCR) and droplet digital PCR (ddPCR) analysis in order to detect viral RNA [58]. The liver, hypophysis, thyroid, spleen, adrenal glands, uterus, or prostate, besides the brain, lungs, heart, and kidneys, were also included in the routine histopathological examination. Upon Hematoxylin–Eosin (H&E) staining, we did not observe any peculiar features that could be related to COVID-19. Moreover, the subjects of the present study did not show any clinical signs related to a possible liver failure or impairment. Therefore, despite the liver being one of the most important organs of the body, we chose to perform the study on the brain, lungs, heart, and kidneys, for which the clinical picture and the routinary H&E staining provided the most interesting results.

In accordance with Italian Law, this research was performed on small portions of biological samples routinely taken during autopsies that had already been examined for diagnostic and/or forensic purposes. The subjects of the study were kept anonymous. The reference law is the authorization n9/2016 of the guarantor of privacy, then replaced by Regulation (EU) 2016/679 of the European Parliament and of the Council. The ABB autopsy and sampling protocol [59] were approved by the Ethics Committee of the University of Pavia on 6 October 2009 (Committee report 3/2009).

### 2.3. Histology and Immunohistochemistry

All sections were compared for morphology using Hematoxylin–Eosin (H&E) staining. Alzheimer’s Disease (AD) severity was assessed on sections immunostained with antibodies against beta-amyloid (4G8, monoclonal antibody, BioLegend San Diego, CA, USA; 1:1000) and phospho-tau (AT8, monoclonal antibody, clone MN1020, Thermo Fisher Scientific Waltham, MA, USA; 1:200) and defined according to Montine’s scheme (low–intermediate–high AD pathology) [60].

To assess SARS-CoV-2 presence, inflammatory infiltrates, and microthrombi, the following anatomical regions were considered: inferior left lung lobe, right kidney, left heart ventricle, frontal lobe (gray–white matter) for the forebrain, and pons for the hindbrain. Antibodies against the following antigens were used: SARS-CoV-2 nucleocapsid (monoclonal antibody, clone B46F, and Invitrogen Waltham, MA USA; 1:100), CD68 (monocytes and activated macrophages: polyclonal antibody and Invitrogen; 1:500—brain microglia:monoclonal antibody clone KP1, and Dako Santa Clara, CA 95051 United States; 1:100), CD3 (T-lymphocytes: monoclonal antibody, clone SP7, and Invitrogen; 1:200—brain: monoclonal antibody, clone F7.2.38, and Dako; 1:50), CD20 (B-lymphocytes: polyclonal antibody and Invitrogen; 1:300—brain: monoclonal antibody, clone L26, and Dako; 1:100), CD42b (activated platelets: monoclonal antibody, clone 42C01, and Invitrogen; 1:100), and GFAP (astrocytes: polyclonal antibody, and Dako Z0334; 1:1000).

The most affected sections were chosen among all of the anatomical regions; successively, low magnification (4×) was used to explore the slide and higher magnifications (10–20×) to investigate the morphological aspects. For each representative slide, 5 areas of 4.7 mm^2^ were evaluated (the 4 corners and the center). In order to characterize the infiltrate, the most affected area was selected for scoring. To grade the reactions, comparable semi-quantitative 4-point scoring systems (0–3) were used. To quantify the presence of lymphocytes and monocyte–macrophages in the different infiltrates of the various tissues, we effectively applied the method described by Matschke and colleagues based on cell counts: 0/4.7 mm^2^ = 0, none; 1–9/4.7 mm^2^ = 1, mild; 10–49/4.7 mm^2^ = 2, moderate; and >49/4.7 mm^2^ = 3, severe [35]. To evaluate the activation of brain microglia, we applied the 0–3 scoring method already consolidated in our laboratory [34]. To quantify the microthrombi, the thrombosed capillaries were counted (0/4.7 mm^2^ = 0, none; 1/4.7 mm^2^ = 1, mild; 2/4.7 mm^2^ = 2, moderate; and ≥3/4.7 mm^2^ = 3, severe). Scores of 0–1 (none-mild) were considered not relevant in all tissues and reactions, while scores of 2–3 represented a moderate to severe pathological alteration. Two neurologists with expertise in neuropathology and two pathologists blinded to the clinical history performed the pathological assessment. Whenever discrepancies between the gradings emerged, the area was reassessed together until an agreement was reached.

### 2.4. Statistical Analysis

All statistical analyses were conducted using R (version 4.2.1; R Core Team; R Foundation, Released 2021, Vienna, Austria). *p*-values of < 0.05 were considered significant. Given the ordinal nature of the scores and the low number of cases, a nonparametric statistical test was used. The T and B lymphocyte sum was considered as a further variable. Friedman’s test with Durbin–Conover pairwise comparison was used to compare the different tissues within subjects (R package PMCMR). Score differences between the cases and controls for each organ were compared using the Mann–Whitney U test.

## 3. Results

### 3.1. General and Clinical Characteristics

The demographic and clinical features of the study participants are shown in Table 1. The nine COVID-19 patients (four females and five males) died 0 to 32 days after diagnosis (mean: 10 days). At death, their mean age was 77.4 (range: 29–94), and the mean post mortem interval was 7 days (range: 3–13). All subjects, except for patient COV2 (a previously healthy young man), had several comorbidities of varying severity, including pulmonary diseases, hypertension, diabetes, obesity, and cancer. None of them had severe heart failure. Six had a history of NCD (four major-NCD and two mild-NCD), five of whom had a clinical course complicated by delirium (three as the first COVID-19 symptom). The other three were cognitively normal. All cases developed severe lymphopenia and typical symptoms (fever–cough–dyspnea), except for the COV2 case, who was asymptomatic and died from hemorrhagic shock due to accidental trauma. Three had sepsis before death and only one was treated in an intensive care unit; however, none of them underwent orotracheal intubation (Table 1). The five cases with non-COVID pneumonia (from the Institute of Legal Medicine) and the five non-COVID ABB controls were matched for age and comorbidities. These subjects died of either of the following: pneumonia–pulmonary failure, heart failure, cachexia due to terminal dementia, or cancer.

### 3.2. Pathological Findings in COVID-19 Cases

The general and specific pathological findings of the COVID-19 cases are summarized in Table 2, and their most representative histological details are displayed in Figure 1.

The lung samples showed: (1) severe capillary congestion, edema, and DAD with cytopathic alterations in type-2 pneumocytes; (2) SARS-CoV-2 positivity in alveolar pneumocytes of five cases, ubiquitous alveolar macrophages, interstitial pneumonia with fibrosis and moderate to severe inflammatory septal infiltrates (mainly T-lymphocytes, present in all cases), and frequent superimposed bacterial infection; and (3) microthrombi and frequent clots inside the vessels in seven cases.

Findings from the kidneys included: (1) congestion and acute glomerular alterations in three cases; (2) SARS-CoV-2 positivity in the tubular epithelium and the capillary endothelium in three cases, and moderate to severe inflammatory infiltrates (T-lymphocytes and B-lymphocytes in the majority of cases, while macrophages were quite rare in all but one case); and (3) microthrombi and clots in two cases.

The heart samples revealed: (1) hypoxic myocytic injuries; (2) sporadic SARS-CoV-2 traces in the endothelial and endocardial cells of one case, perivascular–parenchymal inflammatory infiltrates (prominent only in two cases) characterized by predominant macrophages with no evidence of T-lymphocytes and rare B-lymphocytes; and (3) relevant microthrombi in only one case.

The neuropathological hallmarks were: (1) diffuse cortical edema due to extreme hypoxia with cortical swelling, spongiosis, and severe neuronal rarefaction in the cerebral cortex and hippocampus; (2) very limited traces of SARS-CoV-2 antigens in pontine neurons of one case, perivascular and parenchymal inflammatory infiltrates characterized by the enhancement of CD68-positive amoeboid cells (activated microglia), which are more abundant in the pons (all cases) than in the frontal cortex, with a tendency to nodular aggregation and neuronophagia, and very scant B–T lymphocytes as vascular cuffing or inside few nodules; and (3) frequent microthrombi in the frontal lobe (eight cases) and pons (six cases) with rare ischemic rarefaction of the surrounding tissue.

Although all COVID-19 subjects had a positive pharyngeal swab, none of them expressed positivity for SARS-CoV-2 RNA in the brain using qRT-PCR. Nonetheless, traces of viral RNA were detected in the frontal lobe of almost all cases through ddPCR, a more sensitive technique.

Apart from COV2, all of the other cases had pre-existing or age-related pathologies. The lungs frequently showed emphysema, dystelectasis, and anthracosis. The kidneys showed glomerulosclerosis and arteriolosclerosis. The heart samples presented with myocardiosclerosis, lipofuscin deposits, and fatty infiltrates. The brain samples showed different degrees of atrophy and AD pathology (six cases showing cortical neuritic plaques associated with microglial activation and astrogliosis with reactive astrocytes), and small-vessel disease (SVD), including enlarged perivascular spaces, arteriolosclerosis, myelin loss, hemosiderin leakage, and microbleeds (five cases).

### 3.3. Pathological Findings in Control Non-COVID Cases (n = Five Lungs; n = Five Brains)

The lungs of non-COVID cases showed congestion, edema, and DAD with septal lympho-monocytic infiltrates associated with intra-alveolar fibrinopurulent exudates consisting of neutrophils and macrophages (similar to COVID-19 cases with superimposed bacterial infection). The control brains presented both AD pathology and vascular diseases (SVD and cerebral infarcts). Similar to the COVID-19 cases with AD, the three controls affected by AD showed severe astrogliosis and cortical microglial nodules with a distribution resembling that of neuritic plaques.

### 3.4. Comparison of Pathological Findings

The presence of T-B lymphocytes as a whole (sum of scores) was similar in the lungs and kidneys (Figure 2A; Table 3), albeit with a significantly greater presence of T lymphocytes in the lungs (*p* = 0.020; Figure 2B). The lymphocyte component within the inflammatory infiltrates of lungs and kidneys was clearly predominant over that of other tissues (*p* < 0.001; Figure 2A; Table 2 and Table 3), as well as the presence of the viral antigen, particularly in the type-2 pneumocytes and bronchiolar epithelial cells (Figure 1B,B’). The heart had relevant inflammatory foci in only two cases, characterized by the presence of macrophages (Figure 1O), the substantial absence of lymphocytes, and very rare SARS-CoV-2 traces in the endothelium–endocardium (Figure 1N).

Comparing lung T-B lymphocytes between COVID-19 cases and controls, there was no significant difference (Figure 3A); nonetheless, the T-component was considerably more represented in COVID-19 pneumonia (Figure 1D) (*p* = 0.010; Figure 3B).

As for the brain, the preponderance of microglial activation (innate immunity) over the lymphocytic response (adaptive immunity) was distinct. In the frontal lobe, this phenomenon did not differ between COVID-19 cases and controls; these two groups showed similar levels of microglial activation, probably reflecting an inflammatory boost related to the presence of neurodegeneration (Figure 4A). On the other hand, the significantly greater microglial activation in the pons of COVID-19 cases (Figure 4B), associated with traces of viral antigen (Figure 1T), emerged as a specific topographical phenomenon within the brain (*p* = 0.017; Figure 4B). Microthrombosis assumed a clinico-pathological relevance only in the lungs, with a significant prevalence of pulmonary microthrombi in COVID-19 cases (Figure 5; *p* = 0.023) in comparison with non-COVID pneumonia.

## 4. Discussion

Due to its intrinsic characteristics, SARS-CoV-2 infection is accompanied from the earliest stages by an extreme cytokine outpouring. This so-called “cytokine storm” is a form of severe inflammatory response syndrome due to a hyperactivation of the innate immune system with dysregulated and excessive production of pro-inflammatory cytokines, including IL-6, IL-1, IFN, and TNF-alpha [61]. This type of reaction is specifically related to highly pathogenic beta-CoVs and is frequently observed among patients affected by severe COVID-19. Our results confirm that inflammatory and immune-mediated alterations augment the direct cytopathic damage induced by the virus, causing epithelial and endothelial damage, vascular leakage, and dampening of the T-cell response, accompanied by the over-activation of cells from the macrophage lineage. Hence, the severe form of COVID-19 is a multi-organ disease characterized by a combination of viral invasion, lympho-monocytic infiltration, and clotting alterations with mixed detrimental effects due to acute cytopathic injury, inflammation, microthrombosis, and chronic suffering leading to fibrosis. Further detrimental effects are induced by lung failure causing severe hypoxia in all tissues. The main results to discuss are summarized in the following points. (A) SARS-CoV-2 antigens: both pulmonary and renal tissues present capsid antigens, respectively, in the bronchial epithelia and alveolar pneumocytes, as well as in the tubular epithelial and capillary endothelial cells, but the organ with the heaviest burden of viral antigens is the lung, while their presence is negligible in the heart (rare endothelial cells of one case) and brain (rare pontine neurons of one case), with no evidence of active viral replication; (B) inflammatory infiltrates: lymphocytic presence is most prominent in pulmonary and renal tissues, while the heart and brain display very scant lymphocytes with a clear predominance of the monocyte–macrophage–microglia compartment; (C) comparison between COVID-19 and other types of pneumonia: T-lymphocytes were significantly more represented in the lungs infected by SARS-CoV-2; (D) comparison between COVID-19 and control brains: in the frontal cortex of COVID-19 cases, there was a slight and non-significant microglial boosting that was probably related to pre-existing neurodegeneration (AD pathology), rather than to COVID-19, while microglial hyperactivation was significantly higher in the pons of COVID-19 cases, showing several microglial nodules that appeared to be specifically related to SARS-CoV-2 infection; (E) microthrombosis: while it is a frequent finding across all organs, it appears to represent a specific COVID-19 feature only in the lungs.

This research work has some limitations: (1) the low number of cases of mainly elderly people, who happen to constitute the most affected population, thus representing an interesting standpoint; (2) the study comprised patients from elderly care units who died during the first tumultuous pandemic peak, most of whom were not hospitalized; thus, the serum and blood parameters measured before death were not available in most cases, and the determination of a correlation between blood parameters and pathological changes was not possible; (3) the lack of RNA samples from all tissues apart from the frontal lobe, as most tissue samples from the autopsy were immediately formalin-fixed and paraffin-embedded for safety reasons, making subsequent RNA extraction very difficult in terms of quality and quantity; (4) the predominance of superimposed bacterial infection in the control lungs, making them “not pure” controls for viral pneumonia; however, superimposed bacterial infection was also very frequent in SARS-CoV-2 pneumonia. The strength of this research is the focus on histological differences between organs and tissues of clinically well-documented COVID-19 cases, along with the comparison with lungs and brains from non-COVID matched controls to estimate the specific role of SARS-CoV-2 in determining the pathological changes. Furthermore, we developed a method for histological assessment that considered large areas of tissue to select the most affected zones of an individual organ, which were used for scoring. This approach was proven effective in characterizing and comparing the inflammatory infiltrates.

### 4.1. SARS-CoV-2 Antigens

The parallel comparison between organs confirmed that each tissue interacted differently with SARS-CoV-2, showing interesting similarities and differences. As previously observed by us and other authors [22,24,30], SARS-CoV-2 replicates predominantly, and persists for longer periods, in the alveolar and bronchial epithelium (five cases), where it induces severe cytopathic effects (atypia and death of type-2 pneumocytes). These findings correlate with the abundance of ACE2, TMPRSS2 serine threonine transmembrane protease, and basagin (CD147), which are expressed not only in the alveolar epithelia, but also in the bronchial epithelia. CD147 is now being recognized as a secondary docking site that may increase SARS-CoV-2 virulence and tropism for the upper airways compared with SARS-CoV [62,63].

Similarly, but to a lesser degree, the kidneys showed occasional viral antigens inside the tubular epithelial and capillary endothelial cells. The presence of viral particles in the tubular epithelium has also been reported in other studies [27,64] and may contribute to renal damage. Acute tubular injury is, in fact, often observed in severe COVID-19, and the cause is likely multifactorial; it may result from hypoxia, vascular dysfunctions, severe inflammation, and cytokine release syndrome, along with renal viral tropism, and its consequent direct cytopathic effect on tubular epithelial cells [27,65]. In renal tissues, the co-expression of ACE2 receptors and TMPRSS2 protease has been reported [66,67,68]. This information, coupled with our results, suggests that some viral replication may also take place in the kidneys.

In the heart, SARS-CoV-2 positivity was found only in the endocardium and endothelium of a single case. Even though many have found the presence of ACE2 and TMPRSS2 to be consistent in the heart (especially in people with heart comorbidities), our results suggest that myocardial damage is not imputable to direct viral assault. The virus probably penetrates the heart, but the lack of cytopathic findings and the viral antigen negativity, observed by us as well as by others [31], indicates that the virus does not replicate within the heart, despite the presence of ACE-2 receptors that are, however, mainly expressed by endothelial cells. Further studies are required to elucidate the biological reasons for the absence of active viral replication in the myocardium. It is possible that multiple alternative causes concur with the cardiac damage, among which, invariably, are generalized hypoxia, inflammatory damage, and microangiopathy [20,33].

From a neuropathological standpoint, our data suggest that SARS-CoV-2 slightly penetrates the brain, but does not actively replicate within it. In the brain tissue, ACE2 and TMPRSS2 are very scarcely present [54,69]. Indeed, our results demonstrate very limited traces of viral proteins in a cluster of neurons located in the pons of a single case. The antigen positivity likely results from virions or viral particles ascending from the respiratory and pharyngeal mucosa through the lower cranial nerves. This hypothesis is in line with the findings reported by Matschke et al. [35], who identified SARS-CoV-2 in the lower cranial nerves. SARS-CoV-2 may also be present in the brain through the direct infection of endothelial cells that have a receptor structure favoring direct infection by the virus. Indeed, some authors have reported the sporadic presence of viral antigens within the brain endothelia [45,53,70] and the possible occurrence of endotheliitis [41,42,71]. In our study, viral RNA was detected in minimal quantities in almost all brains using ddPCR, which is a very sensitive method capable of amplifying fragments of the viral genome originating from the blood [58].

Our data suggest that SARS-CoV-2 causes an acute infection with progressive “cleaning” of the virus from the affected tissues. In particular, the virus was not detectable in four out of nine of the COVID-19 lungs, while a significant inflammatory infiltrate persisted in all cases. Along with other studies [54,72], our data indicate that, similar to SARS-CoV and MERS-CoV, SARS-CoV-2 does not produce a persistent infection. Eventually, the virus is cleared from the tissues it infects; nonetheless, the consequences of the disease may last longer, resulting in persistent symptoms lasting weeks to months after the acute phase. Hence, an accurate analysis of the inflammatory phenomena that accompany SARS-CoV-2 infection is important.

### 4.2. Inflammatory Infiltrates

Examining the inflammatory infiltrates more closely, it is evident how the lymphocytic component is essentially more prominent in the lungs, less prevalent, but still relevant, in the kidneys, and negligible, if not absent, in the heart and brain (Figure 2A). In particular, the sum of lymphocytes showed the highest scores in the pulmonary and renal tissues, where it tended to be superimposable (Figure 2A). Although the lungs and kidneys seemed to behave similarly, the lungs actively reacted to the massive viral invasion and replication, which was not so evident in the renal tissue. Indeed, T-lymphocytes were significantly more prominent inside the pulmonary infiltrates (Figure 2B). This was probably a specific adaptive immune response against the intense viral replication, as demonstrated by the presence of the highest T-lymphocyte scores in association with the abiding positivity for SARS-CoV-2 (Table 2).

Contrarily, the heart and brain, which were less directly affected by the infection, displayed a scant lymphocytic presence, and the preponderance of macrophages and microglial cells, respectively, as a non-specific immune response (innate immunity) to antigenic perturbation and immune-complex formation. In the heart samples, we observed moderate macrophage infiltration in two cases (Table 2). Such findings confirm the possibility of a macrophagic inflammatory response in the heart that has also been reported by others [32]. It is not yet clear whether this finding may represent a pathophysiological basis for the occasional reported cases of myocarditis and/or pericarditis in the literature [73,74,75]. Overall, pathological reports regarding the heart have yielded inconsistent results, and the mechanism of myocardial injury is probably multifactorial.

### 4.3. Comparison between COVID-19 and Other Types of Pneumonia

It is noteworthy that COVID-19 and non-COVID types of pneumonia share the same pathology with congestion, edema, DAD with septal lympho-monocytic infiltrates, and intra-alveolar exudate. A number of scientific articles have recently been published regarding the pulmonary features of COVID-19, describing several clinical, radiologic, and autopsy findings that closely resemble those seen in SARS and MERS cases, and also in other types of viral pneumonia, such as H1N1 flu cases [25]. These observations are consistent with ours; in particular, DAD emerges as a common key pathophysiological mechanism. Such findings suggest that the pathophysiology of alveolar damage in SARS-CoV-2 infection is the same as that of other known causes of acute respiratory distress syndrome (ARDS). Nonetheless, our results outline the prominent T-lymphocyte response precisely in the site of the greatest viral replication. Indeed, even with a similar inflammatory infiltrate (sum of B and T lymphocytes not significantly different; Figure 3A), a significantly greater presence of T lymphocytes was observed in COVID-19 lungs compared with the control ones (Figure 3B). The relevant presence of T lymphocytes underlines the role of these cells in the specific response where viral replication is particularly active. We can assume that such a response may be common to other types of pneumonia with purely viral etiology.

### 4.4. Comparison between COVID-19 and Control Brains

In the analysis of the central nervous system, we focused on the frontal lobe and the pons, which are two important areas of the forebrain and hindbrain, respectively. Brain pathology is characterized by the hyperactivation of microglia that exhibit amoeboid morphology and phagocytic properties, which have also been described by others [37,76]. In turn, microglial amoeboid cells tend to agglomerate into micronodules. When comparing the frontal lobe of people with COVID-19 with that of matched controls, we found similar features (Figure 4A). It should be considered that they were almost always elderly people with cognitive problems affected by some degree of pre-existing vascular or degenerative pathologies that, per se, induced cortical hypoxia and inflammatory changes. Indeed, we have already reported that, in those cases with dementia, the distribution of the inflammatory nodules closely paralleled that of amyloid plaques, regardless of SARS-CoV-2 infection [34]. Instead, regardless of the cognitive state, microglial hyperactivation was significantly more intense in the pontine structures of COVID-19 cases compared with controls (Figure 4B). This phenomenon, also observed by others [35,76], appears to be specific to the “COVID-19 encephalopathy” and may be activated by viral debris and isolated virions originating from the tracheobronchial and oropharyngeal mucosa through the lower cranial nerves [35]. Moreover, microglial activation in the brain is probably enhanced by infection-induced cytokine release and blood–brain barrier damage due to immune complex formation [76]. The topography of inflammatory lesions during SARS-CoV-2 infection represents the neuropathological basis of “COVID-19 encephalopathy”, which clinically presents as behavioral changes, lethargy, vegetative and autonomic dysfunctions, and absence of hypoxic drive [34,76,77].

The overall impact triggered by hypoxia and inflammation may accelerate neurodegeneration, causing the so-called “brain fog” occurring after the acute phase of the disease [13]. These phenomena may be enhanced in the elderly by the presence of a pre-existing degenerative burden, which causes microglial priming that, in turn, results in a more intense inflammatory response, favored by “immunological senescence” (lowered adaptive immunity with lower lymphocytic specific response) and “inflammaging” (age-related hyperactivation of innate immunity leading to an excessive non-specific inflammatory reaction) [78,79,80]. Furthermore, transcriptomics studies conducted by us and others confirm the pivotal role of hypoxia and persistent microglial activation in the brain, by demonstrating transcriptional changes in the genes involved in the hypoxic response and the modulation of several microglial functions, including migration and phagocytic induction [58,81]. Where there was no active viral replication, a non-specific macrophage–microglial response predominated, which, in elderly subjects, can be favored by the neurodegenerative load, hence demonstrating the importance of a comparison that included patients with AD (the most common neurodegenerative disease).

Although several authors have proposed theories regarding a possible neurotropism of SARS-CoV-2 and its possible persistence in the CNS causing long-term consequences [51,53,82], in our cases, the brain showed scant lymphocytic infiltration and very limited traces of SARS-CoV-2 antigens with no associated evidence of viral replication and encephalitis. The pathological features we have shown are quite different from those of viral encephalitis caused by neurotropic viruses [83], in which the presence of abundant viral antigens, abundant lymphocyte infiltrate, and direct cytopathic effects are observed, as well as that occurring in the lung. From these observations and literature analysis, it is inferred that SARS-CoV-2 is probably not a neurotropic virus and, importantly, there is no evidence proving its persistence within the brain after acute infection, at least in most cases.

### 4.5. Microthrombosis

Regarding the occurrence of thrombosis and microthrombosis, it should be considered that these phenomena, due to both clotting and endothelial alterations, are described in more than half of COVID-19, SARS, and MERS cases, while they are fairly less common in pneumonia caused by A/H1N1. Such findings suggest that thrombotic complications may be more specifically correlated with beta-CoVs than flu viruses [25,84,85]. In our study, we did not observe gross abnormalities, thrombosis of the large vessels, or infarcts; these phenomena have been reported by others [86,87,88] and are probably related to a protracted clinical course, which was not the case in our series. Nonetheless, we noted frequent microthrombosis in all organs. In particular, this phenomenon was significantly more prominent in COVID-19 lungs compared with control lungs (Figure 5) affected by non-COVID pneumonia. This confirms that microthrombosis is an event specific to SARS-CoV-2 infection. The pathology of such a phenomenon may be explained by Virchow’s triad: (1) the endothelial dysfunction and endothelial damage due to viral tropism and endotheliitis [7]; (2) the hypercoagulability state and increased blood viscosity due to the release of damage-associated molecular patterns (DAMPS) and to the higher load of cytokines, immunoglobulins, and immune complexes traveling within the blood [89,90]; and (3) the prolonged stasis that may result from immobilization during hospitalization. Microthrombosis in the lungs is an event specific to SARS-CoV-2 pneumonia and contributes to the clinical severity, lung failure, and mortality.

Our results confirm that thrombosis inside small pulmonary vessels (Figure 1F) is a COVID-specific phenomenon. The same cannot be stated for the small vessels of the brain, in which microthrombi are present both in cases and in controls, and are probably more related to prolonged agony, co-morbidities, and post-mortem phenomena, rather than to SARS-CoV-2 infection. Indeed, the only young case (COV2) with rapid death and no concomitant pathologies had little or no presence of microthrombi in all organs, suggesting a possible contribution of agony length, age, and co-morbidities to platelet aggregation inside small vessels.

## 5. Conclusions

We can conclude that: (1) viral replication appears to be active and have a direct pathogenetic role in the lungs and, to a lesser degree, in the kidneys; (2) the type of infiltrate depends on the relationship that the virus establishes with the tissue; in particular, the more active viral replication is, the more T-lymphocytes are present, while a macrophage–microglial response predominates where there is no evidence of active viral replication; and (3) the most specific COVID-19 pathological features consist of an abundance of T-lymphocytes and microthrombosis at the pulmonary level, and relevant microglial hyperactivation in the brainstem.

A careful examination of the pathological pictures present in the various tissues is the basis for understanding acute and long-term symptoms. Overall, our findings suggest that tissue damage in the lungs and kidneys may be caused by a direct viral cytopathic effect along with inflammation-mediated mechanisms. On the other hand, the heart and brain may be damaged mainly by an abnormal and persistent inflammation. Additionally, pre-existing pathologies (e.g., neurodegeneration) and COVID-19’s clinical course (e.g., presence of critical illness, hemodynamic instability, hypoxia, and sepsis) affect the clinicopathological pictures. The presence of sequelae across all organs appears to be the result of a combination of the aforementioned factors. It is probable that a complete recovery from COVID-19 requires the termination of both viral infection and the associated inflammation, which can take many months.

The biologically detrimental effects of SARS-CoV-2 infection and related inflammatory changes are at least partially reversible. A deeper understanding of these phenomena is important to improve the management of COVID-19 patients, also after the acute phase. During the post-acute phase of the disease, rehabilitative interventions, such as physical activity, cognitive training, and psychosocial support, should be provided as soon as possible to restore previous functional performances.

## Figures and Tables

**Figure 1 cells-11-03124-f001:**
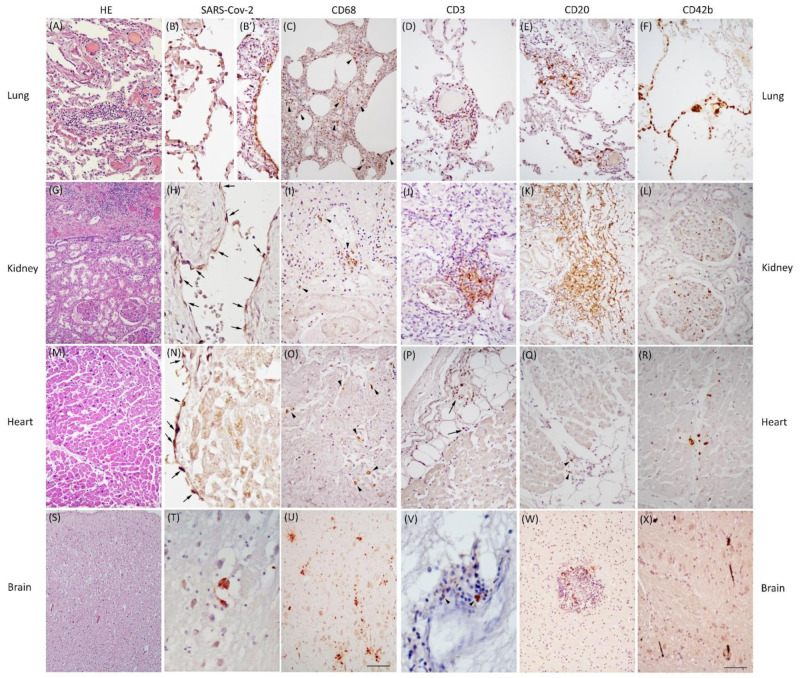
Pathological features of COVID-19. In the lung, H&E revealed severe congestion, diffuse alveolar damage, and interstitial pneumonia presenting as septal infiltrate shown in the middle of the image (**A**); SARS-CoV-2 positivity is evident in alveolar pneumocytes showing cytopathic features (**B**) and in the bronchiolar ciliated epithelium (**B’**); diffuse interstitial macrophages detected by CD68 antibody (**C**, arrowheads); T-lymphocyte (**D**) and B-lymphocyte (**E**) infiltrates are revealed by CD3 and CD20 reactions, respectively; CD42b marks several microthrombi in capillary and interstitial vessels characterized by a “rosary crown” feature (**F**). In the kidney, H&E labeled acute glomerular alterations in the lower part of the image and inflammatory infiltrates at the top (**G**); SARS-CoV-2 immunoreactivity detectable in some vascular endothelial cells (**H**, arrows); occasional foci of macrophages detected by the CD68 antibody (**I**, arrowheads); as in the lung, T-lymphocyte (**J**) and B-lymphocyte (**K**) infiltrates are revealed by CD3 and CD20 reactions, respectively; occasional microthrombi in the glomerular capillary are marked by CD42b antibody (**L**). In the heart, H&E stained parenchymal dissociation and myocyte vacuolization in the upper part of the image (**M**); rare SARS-CoV-2 traces observed in the endocardium (**N**, arrows); occasional foci of interstitial macrophages labeled by the CD68 antibody (**O**, arrowheads), rare subepicardial T-lymphocyte (**P**, arrows) and B-lymphocyte (**Q**, arrowheads) infiltrates revealed by CD3 and CD20 reactions in the upper and lower parts of the images, respectively; CD42b antibody marks focal and sporadic capillary microthrombi (**R**). In the brain, H&E revealed diffuse neuronal loss and cortical edema characterized by spongiosis (**S**); very rare SARS-CoV-2-positive cells detected in the pons (**T**); amoeboid microglial cells and several microglial nodules identified by the CD68 antibody, mainly in the brainstem (**U**); rare T and B lymphocytes are observed in the perivascular spaces (**V**) and in some nodules (**W**); frequent capillary microthrombi are observed in the brainstem (**X**). Scale bars: 230 μm (**S**); 162 μm (**G**); 140 μm (**C**,**U**); 75 μm (**A**,**E**,**F**,**K**,**M**,**P**,**W**); 64 μm (**D**,**I**,**O**,**Q**,**R**,**X**); 60 μm (**J**); 52 μm (**L**); 44 μm (**B’**); 39 μm (**B**); and 30 μm (**H**,**N**,**V**,**T**).

**Figure 2 cells-11-03124-f002:**
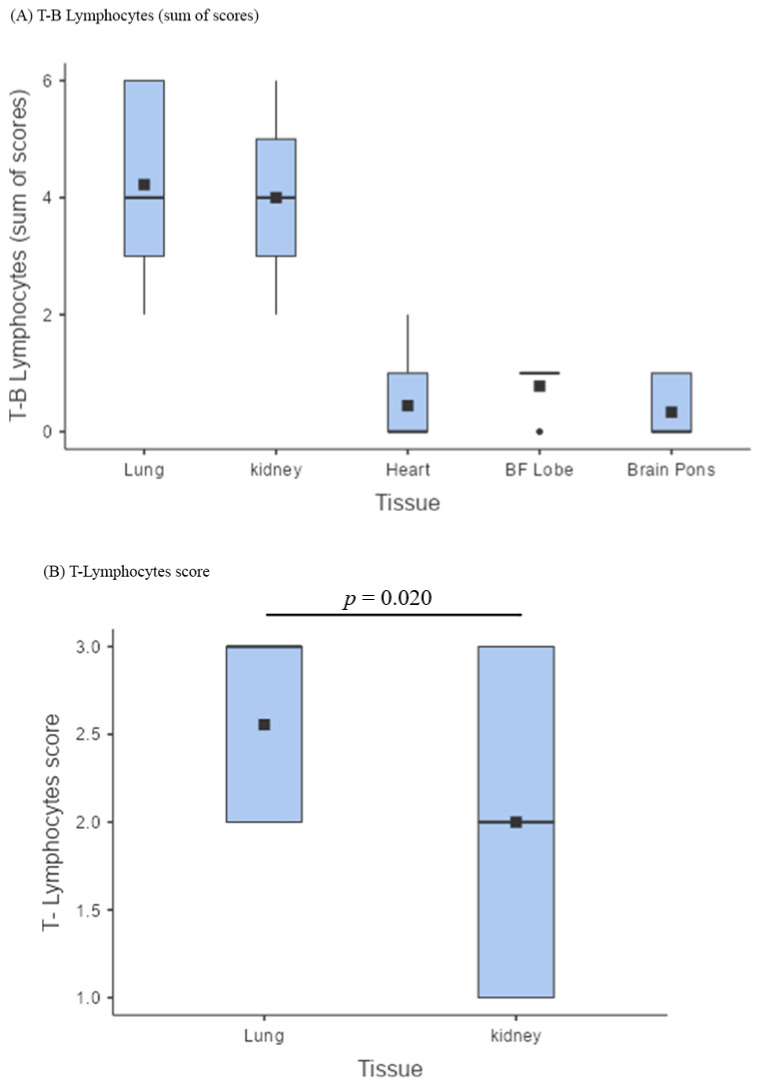
Lymphocytic infiltrates. (**A**) Box plot comparing the sum of the B and T lymphocyte scores across lungs, kidneys, heart, brain frontal lobe (BF), and pons (BP); the lymphocyte component within the inflammatory infiltrates of lungs and kidneys was clearly predominant over that of other tissues (*p* < 0.001); (**B**) box plot showing a comparison between T lymphocyte scores in the lungs and in the kidneys; The presence of T-B lymphocytes as a whole (sum of scores) was similar in the lungs and kidneys, albeit with a significantly greater presence of T lymphocytes in the lungs (*p* = 0.020).

**Figure 3 cells-11-03124-f003:**
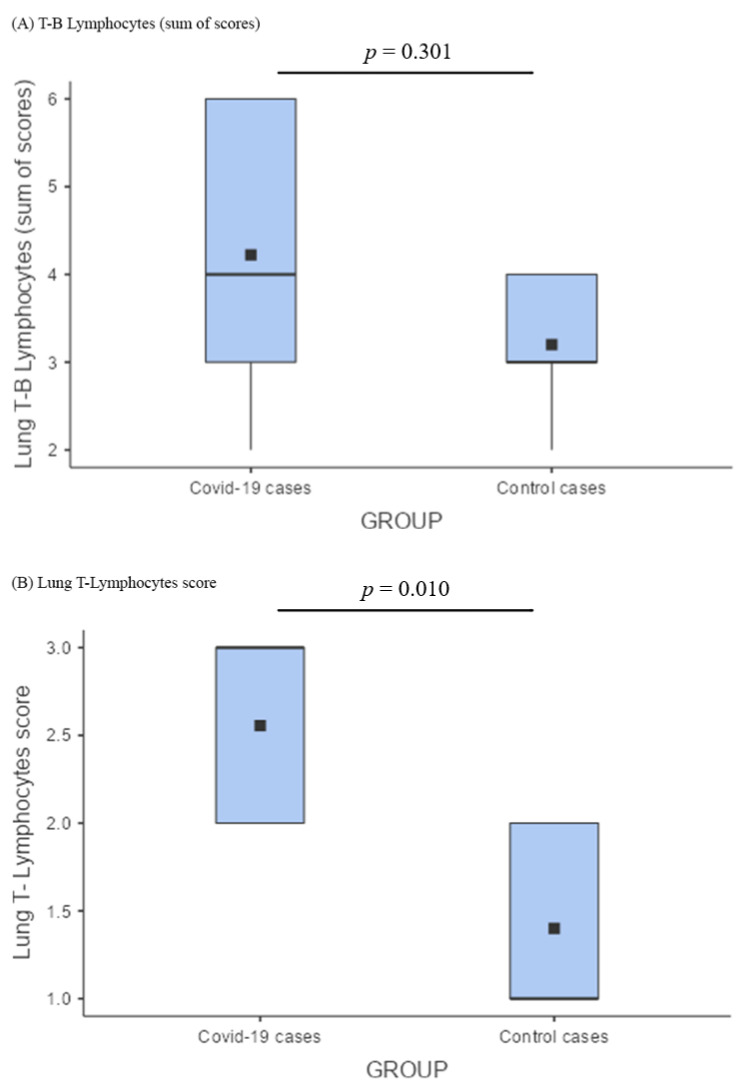
Comparison of lymphocytic infiltrates among the lungs of cases and controls. (**A**) Box plot describing the comparison between the lung B–T lymphocyte of COVID-19 cases versus control cases; lung T-B lymphocytes (sum of scores) did not show any significant difference; (**B**) lung T lymphocyte score comparison among COVID-19 cases and control cases according to a box plot demonstrated that the T-component was considerably more represented in COVID-19 pneumonia (*p* = 0.010).

**Figure 4 cells-11-03124-f004:**
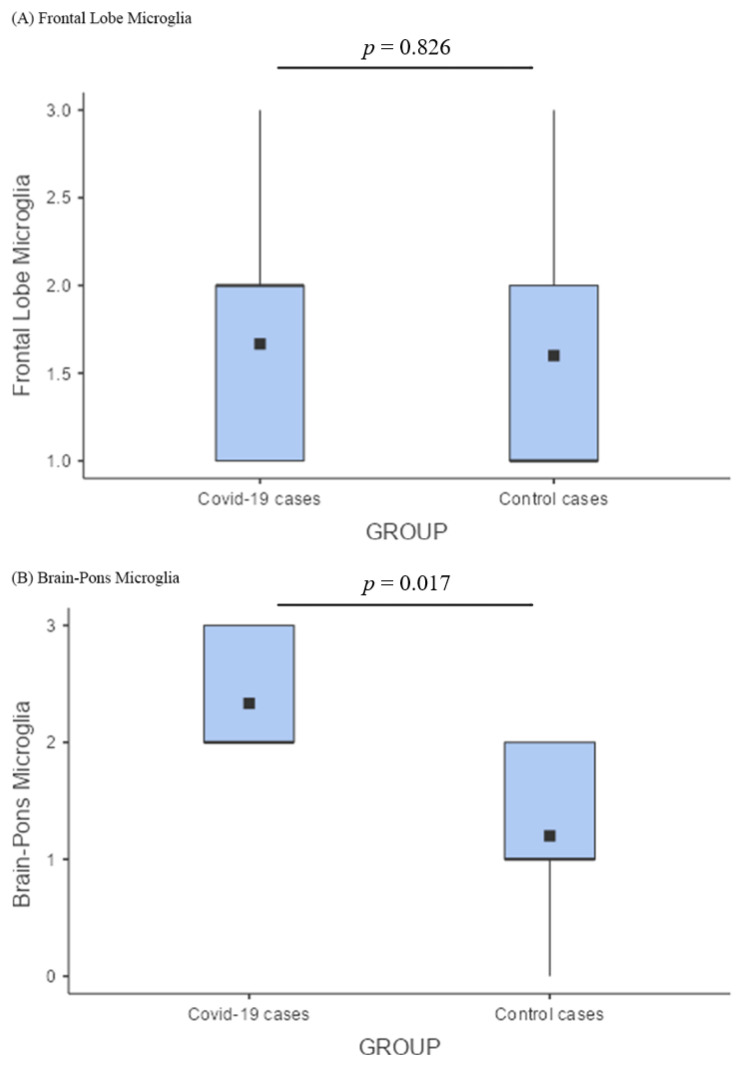
Comparison of microglial activation between the brains of cases and controls. (**A**) In the frontal lobe, microglial activation (innate immunity) did not differ between COVID-19 cases and controls; (**B**) Comparison of pontine microglia between COVID-19 cases and control cases by a box plot showed a significantly greater microglial activation in the pons of COVID-19 cases (*p* = 0.017).

**Figure 5 cells-11-03124-f005:**
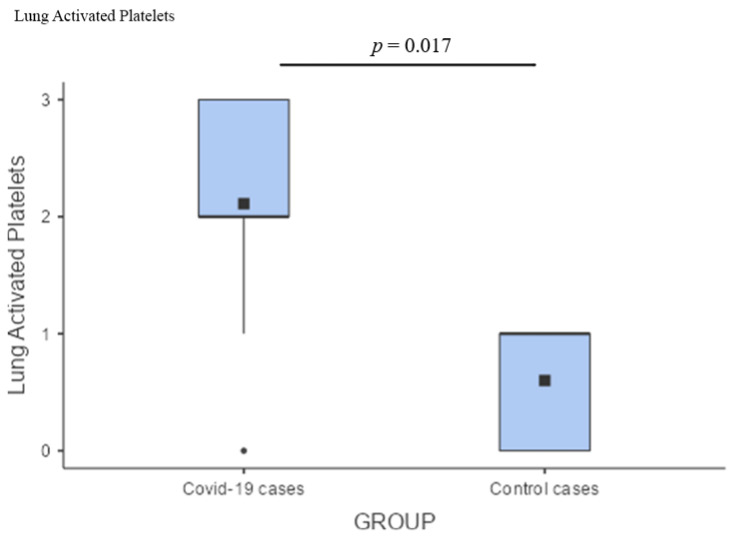
Box plot comparing activated platelets between the lungs of cases and controls demonstrating a predominance of pulmonary microthrombi in COVID-19 cases (*p* = 0.023).

**Table 1 cells-11-03124-t001:** General and clinical information.

Code	PMD (Days/Hours)	General and Clinical Features					
		Sex	Age (y/o)	Anamnesis; Cause of Death	NCD	DEL	SEP
**COV2**	7 d	M	29	NR; hemorrhagic shock	no	no	no
**COV4**	5 d	M	67	Obesity, HTN, and CVD; CIP and respiratory failure	no	no	yes
**COV6**	11 d	F	90	HTN and COPD; respiratory failure	no	no	no
**COV3**	7 d	M	87	T2D and CVD; respiratory failure	Mild (VCI)	Hyper/Hypo (early onset)	no
**COV10**	7 d	M	81	AF and paraparesis (previous GBS); respiratory failure	Mild (VCI)	Hypo (late onset)	yes
**COV1**	7 d	F	74	NR; respiratory failure	Major (AD)	Hyper (early onset)	no
**COV5**	3 d	F	94	T2D, HTN, CVD, and AF; multiorgan failure	Major (AD + VaD)	Hypo (early onset)	yes
**COV8**	13 d	F	83	HTN; respiratory failure	Major (AD)	no	no
**COV9**	6 d	M	92	HTN and cerebrovascular disease; respiratory failure	Major (AD + VaD)	Hyper/Hypo (late onset)	no
**L1**	4 d	F	76	AF; multiorgan failure 7 days after head trauma	no	no	no
**L2**	5 d	M	92	HTN CVD, and cerebrovascular disease; respiratory failure	Major (VaD)	Hypo (late onset)	no
**L3**	6 d	F	60	CVD; multiorgan failure 3 days after head trauma	no	no	no
**L4**	4 d	M	62	HTN; respiratory failure	no	no	yes
**L5**	8 d	M	74	HTN and COPD; multiorgan failure 10 days after intestinal perforation	Major (AD + VaD)	Hypo (late onset)	yes
**B1**	3 h	M	79	T2D; liver cancer	no	no	no
**B2**	8 h	M	79	HTN, CVD, and cerebrovascular disease; cachexia	Mild (VCI) and hemiparesis	no	no
**B3**	16 h	F	83	HTN, CVD, and cerebrovascular disease; CHF	Major (AD + VaD)	no	no
**B4**	15 h	F	85	CVD and cerebrovascular disease; CHF	Major (AD + VaD)	Hyper (prev. ep)	no
**B5**	15 h	F	89	HTN, COPD, and cerebrovascular disease; cachexia	Major (AD)	Hyper (prev. ep)	no

Note: COVID-19 cases are labeled as ‘COV’, control cases are identified as ‘B’ (Brain Bank) for brain and L (lung control) for lungs; PMD was measured in days (d) or hours (h). Abbreviations (in alphabetical order): AD, Alzheimer’s disease; AF, atrial fibrillation; CHF, chronic heart failure; COPD, chronic obstructive pulmonary disease; CVD, cardiovascular disease; DEL, delirium; GBS, Guillain Barre Syndrome; HTN, hypertension; n/a, not available; NCD, neurocognitive disorder; NR, nothing relevant in medical history; PMD, post mortem delay; prev. ep., previous episode; SEP, sepsis; T2D, type 2 diabetes; VaD, vascular dementia; VCI, mild vascular cognitive impairment.

**Table 2 cells-11-03124-t002:** Pathological data. *Acute alterations*: AE, acute emphysema; AGA, acute glomerular alterations; AH, alveolar hemorrhage; BS, bacterial superimposition; C, congestion; DAD, diffuse alveolar damage; E, edema; HA, hypoxic alterations (congestion, edema, spongiosis, and neuronal loss); IF, inflammatory foci; MA, myocyte alterations (parenchymal dissociation and/or wavy fibers); MV, myocyte vacuolization; RA, reactive astrocytes. *Chronic alterations*: A, anthracosis; AD, Alzheimer’s disease; AS, atherosclerosis; CE, chronic emphysema; CH, cardiomyocyte hypertrophy; F, fibrosis; GS, glomerulosclerosis; SVD, small-vessel disease. *Markers:* AP and BL activated platelets (CD42b) and B-lymphocytes (CD20); M, macrophage/microglia (CD68); TL, T-lymphocytes (CD3); VA, viral (SARS-CoV-2) antigen; Vd, viral droplet digital polymerase chain reaction (ddPCR); Vr, viral real-time polymerase chain reaction (RT-PCR). n/a, not available; present, +; absent, -. COVID-19 cases are labeled as ‘COV’, control cases are identified as ‘B’ (Brain Bank) for brain and L (lung control) for lungs

	General Pathological Features				Specific Pathological Features																										
	Lung	Kidney	Heart	Brain	Lung					Kidney					Heart					Brain-Frontal Lobe							Brain-Pons				
CASE	(Acute; chronic findings)	(Acute; chronic findings)	(Acute; chronic findings)	(Acute; chronic findings)	M	TL	BL	AP	VA	M	TL	BL	AP	VA	M	TL	BL	AP	VA	M	TL	BL	AP	VA	Vr	Vd	M	TL	BL	AP	VA
COV2	Mild C, AE, interstitial- subpleural IF; F, CE, A	Severe C	MA	HA; no AD, no SVD, mild gliosis-RA					+					-					-					-	n/a	n/a					-
COV4	Severe C, interstitial-subpleural AH, AE, E, DAD, BS, IF; CE, A	Severe C, cortical-medullary IF; GS	MV, BS, MA, Subepicardial IF; F, CH	HA; no AD, SVD, mild gliosis					-					-					-					-	-	+					-
COV6	AE, E, interstitial IF; F, A	Cortical IF; F	Mild C	HA; low AD, mild gliosis					-					-					-					-	-	+					-
COV3	Severe C, AE, DAD, interstitial IF; AS, CE, A	Severe C, AGA, cortical-medullary IF; F, AS, GS	MV, MA; F, AS	HA; low AD, SVD, mild gliosis					+					-					-					-	-	+					+
COV10	Severe C, AE, BS, interstitial IF; AS	Severe C, AGA; F	MA, interstitial-subepicardial IF; F	HA; no AD, SVD, perivascular gliosis					+					+					+					-	-	+					-
COV1	C, AE, AH, E, DAD, BS, interstitial IF; F, CE	Cortical IF; F, GS	MA, BS; F	HA; high AD, perivascular gliosis					+					-					-					-	-	-					-
COV5	Severe C, AH, AE, E, DAD, BS, interstitial IF; A, CE	Severe C, cortical-IF, AGA; GS	MV, MA, subepicardial IF; F, CH	HA; intermediate AD, SVD, mild gliosis-RA					+					+					-					-	-	+					-
COV8	Severe C, interstitial IF; CE, A	Cortical-medullary IF; AS	F	HA; intermediate AD, perivascular gliosis-RA					-					-					-					-	-	+					-
COV9	Severe C, AH, interstitial IF; CE, A	AS	MA, subepicardial IF; F, AS	HA; intermediate AD, SVD, perivascular gliosis-RA					-					+					-					-	-	+					-
C:L1 / B1	Mild C, E, DAD, interstitial IF; F, A	n/a	n/a	No AD, no SVD					-	n/a	n/a	n/a	n/a	n/a	n/a	n/a	n/a	n/a	n/a					-	-	-					n/a
C:L2 / B2	AE, interstitial-subpleural IF; F, CE, A	n/a	n/a	No AD, stroke and SVD					-	n/a	n/a	n/a	n/a	n/a	n/a	n/a	n/a	n/a	n/a					n/a	n/a	n/a					n/a
C:L3 / B3	Mild C, interstitial IF; E, DAD, BS	n/a	n/a	Intermediate AD, stroke, severe gliosis					-	n/a	n/a	n/a	n/a	n/a	n/a	n/a	n/a	n/a	n/a					n/a	n/a	n/a					n/a
C:L4 / B4	Severe C, AE, BS, interstitial-subpleural IF; A	n/a	n/a	High AD, SVD, severe gliosis					-	n/a	n/a	n/a	n/a	n/a	n/a	n/a	n/a	n/a	n/a					n/a	-	-					n/a
C:L5 / B5	Mild C, E, BS, interstitial IF; F, CE	n/a	n/a	High AD, severe gliosis					-	n/a	n/a	n/a	n/a	n/a	n/a	n/a	n/a	n/a	n/a					n/a	n/a	n/a					n/a

Gradings: 
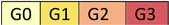
.

**Table 3 cells-11-03124-t003:** Summary of statistical analyses.

	χ^2^	*p*-Value
Friedman rank sum test	31.8	<0.001
Pairwise comparisons (Durbin–Conover)	t	*p*-value
Sum of T-B Lymphocytes Lungs vs. Kidneys	0.438	0.665
Sum of T-B Lymphocytes Lungs vs. Heart	10.506	<0.001
Sum of T-B Lymphocytes Lungs vs. Brain frontal lobe	8.536	<0.001
Sum of T-B Lymphocytes Lungs vs. Brain pons	11.162	<0.001
Sum of T-B Lymphocytes Kidneys vs. Heart	10.068	<0.001
Sum of T-B Lymphocytes Kidneys vs. Brain frontal lobe	8.098	<0.001
Sum of T-B Lymphocytes Kidneys vs. Brain frontal pons	10.725	<0.001

## Data Availability

The dataset of this research has been deposited in the official computer archive of the Golgi-Cenci Foundation, and it is available upon request.

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
