# Peer review of "COVID-19 Pathology in the Lung, Kidney, Heart and Brain: The Different Roles of T-Cells, Macrophages, and Microthrombosis"

_cells, 2022, doi:10.3390/cells11193124_

Round 1

Reviewer 1 Report

The article describes COVID-19 pathology in the lung, kidney, heart and brain based on findings obtained from examination of autopsy samples.

Considering the journal's quality, the article must be improved with additional experiments and a comparison of medical records of COVID-19-positive patients who had recovered from the cytokine storm.

 Comments: 

1. Explain the reasons behind the assessment of autopsy samples for Alzheimer's disease pathology?

There is no mention of Alzheimer's disease and its relation to COVID-19 pathology, the role of T cells, macrophages etc.,...

2. Most importantly, a comparison of medical records for COVID-19 biochemical or radiological changes of COVID-19 recovered patients who had cytokine storm with the current findings must be made. Though pathological changes at the tissue levels cannot be compared, correlation with serum changes (lymphocyte count, COVID-19 markers, serum IL-6, CRP, Ferritin, LDH, serum markers specific to heart, kidney, lung, and brain pathology, if done) may help those recovered patients to take precautionary or preventive measures.

3. Authors must describe reasons why in the heart and the brain viral load was less despite these tissues having ACE2 receptors. Are the pathological changes observed in these organs a consequence of complications of pathological changes in the lung and not a direct effect of the viral load?

4. In the Summary, the authors must clearly explain how these findings have a take-home message for the people who survived COVID-19.

 5. What could be the reasons for the different roles of T-cells, macrophages, and microthrombosis?

6. Authors may explain interactions between Alzheimer's pathology Vs COVID-19 in teh context of roles of T- cells, macrophages and microthrombosis. Alternatively, authors may take out the Alzheimer's reference.

Author Response

We are very grateful to the reviewer for his time, and for providing constructive and helpful advice to improve our manuscript. Here are our answers to all the questions raised by the reviewer:

REV 1

The article describes COVID-19 pathology in the lung, kidney, heart and brain based on findings obtained from examination of autopsy samples.

Considering the journal's quality, the article must be improved with additional experiments and a comparison of medical records of COVID-19-positive patients who had recovered from the cytokine storm.

Comments: 

  1. Explain the reasons behind the assessment of autopsy samples for Alzheimer's disease pathology?

There is no mention of Alzheimer's disease and its relation to COVID-19 pathology, the role of T cells, macrophages etc.,...

Pathologies of COVID-19 and AD are both present and interrelated in our case series and we believe that clinico-pathological manifestation of COVID-19 in the elderly are highly influenced by pre-existing neurodegenerative burden. According to your suggestion we added in the introduction the following sentence:

“In this regard, it should be considered that elderly population is the most affected by the severe form of COVID-19, and many patients had pre-existing neurocognitive disorders; thus, brain inflammatory changes and consequent neurological manifestations may be greatly influenced by the presence of the microglial “priming” due to neurodegeneration”.

However, we already investigated the role of T-cells and macrophages in a previous study quoted (n. 34) in the bibliography (Poloni, Brain Pathol 2021).

  1. Most importantly, a comparison of medical records for COVID-19 biochemical or radiological changes of COVID-19 recovered patients who had cytokine storm with the current findings must be made. Though pathological changes at the tissue levels cannot be compared, correlation with serum changes (lymphocyte count, COVID-19 markers, serum IL-6, CRP, Ferritin, LDH, serum markers specific to heart, kidney, lung, and brain pathology, if done) may help those recovered patients to take precautionary or preventive measures.

The study comprises patients from elderly care units, who died during the first tumultuous pandemic peak, and the most of them were not hospitalized. Medical records essentially contained clinical information; thus, serum and blood parameters measured before death were not available in most cases; the autopsies were required for forensic reasons, and our work is mainly focused on pathology. We added in the limitations (see discussion) the following sentence:

“… the study comprises patients from elderly care units, who died during the first tumultuous pandemic peak, and most of them were not hospitalized; thus, serum and blood parameters measured before death were not available in most cases, and a correlation between blood parameters and pathological changes was not possible”

  1. Authors must describe reasons why in the heart and the brain viral load was less despite these tissues having ACE2 receptors. Are the pathological changes observed in these organs a consequence of complications of pathological changes in the lung and not a direct effect of the viral load?

ACE-2 in the brain is present mainly in the endothelium and very scarcely on glial cells and neurons, see citation (51; Iadecola Cell 2020); furthermore, we have already mentioned the presence of hypoxic damage due to lung failure. As for the heart, the paucity of SARS-CoV-2 presence, despite the presence of ACE-2 receptors, is an important observation that requires further studies to elucidate the biological reasons. We added this point to the discussion:

“…., despite the presence of ACE-2 receptors which, however, are mainly expressed by endothelial cells. Further studies are required to elucidate the biological reasons for the absence of active viral replication in the myocardium.”

  1. In the Summary, the authors must clearly explain how these findings have a take-home message for the people who survived COVID-19.

Because of the potential reversibility of pathological inflammatory changes, the take home message may be quite optimistic for survivors and we emphasize the importance of rehabilitation interventions; please, see the conclusions:

“The biological detrimental effects of SARS-CoV-2 infection and related inflammatory changes are at least partially reversible. A deeper understanding of these phenomena is important to improve the management COVID-19 patients, also after the acute phase. During the post-acute phase of the disease, rehabilitative interventions, such as physical activity, cognitive training and psychosocial support, should be provided as soon as possible to restore previous functional performances.”

  1. What could be the reasons for the different roles of T-cells, macrophages, and microthrombosis?

T-cells are activated only in the tissue where the virus replicates as a specific response. Macrophages and microglia largely prevail in heart and brain, where viral presence is scant, with almost complete absence of T-cell response; micro-thrombosis appears to a typical feature of CoVs pneumonia and contributes to the clinical severity and mortality. We have extensively addressed these topics in the discussion; to better explain the role of microthrombosis in the lung, we added the following sentence in the discussion:

“Microthrombosis in the lung is an event specific to SARS-CoV-2 pneumonia and contributes to the clinical severity, lung failure and mortality.”

  1. Authors may explain interactions between Alzheimer's pathology Vs COVID-19 in the context of roles of T- cells, macrophages and microthrombosis. Alternatively, authors may take out the Alzheimer's reference.

Our COVID-19 case series comprises 6 cases with some degree of AD pathology; microglial activation state is influenced by degenerative burden, and the information about AD pathology is important to judge the specific role of microglia in COVID-19. Moreover, the matched control brains have to include AD cases for comparison. Thus, we studied all brains testing for AD, in order to investigate the role of microglia. To clarify this point, we have inserted the following sentences at the end of the introduction and in the methodological part respectively:

“3) to emphasize the pathological features specific for SARS-CoV-2 infection through a comparison between COVID-19 and non-COVID lungs, and between COVID-19 brains with and without neurodegenerative burden (i.e., Alzheimer’s disease – AD pathology) and non-COVID brains with and without AD pathology;…”

“Due to the presence of cognitive disturbances in over half of the cases, and the fact that these disturbances worsened during COVID-19, we chose to also evaluate the presence of AD neuropathology”

Reviewer 2 Report

The introduction (lines 40-72) of the article should be written more fluently. So I call it the "inverted pyramid method". It is necessary to reduce the subject from the general to the specific. The article should begin with the epidemiology of the disease and then evolve towards the relevant topic. You can find the method I want to explain in the following two articles. You can benefit from the introduction of these two articles.

Sahin TT, Akbulut S, Yilmaz S. COVID-19 pandemic: Its impact on liver disease and liver transplantation. WorldJ Gastroenterol. 2020 Jun 14;26(22):2987-2999

BaÅŸkıran A, Akbulut S, Åžahin TT, Tunçer A, Kaplan K, Bayındır Y, Yılmaz S. Coronavirus Precautions: Experience of High Volume Liver Transplant Institute. Turk J Gastroenterol. 2022 Feb;33(2):145-152

In order for the readers to focus on the article, the "Study design, setting, participants and clinical data" section should be written in a more understandable and clear way.

The authors stated that the aims of this study are (i) to describe how the organs such as lungs, kidney, heart, and brain are involved and how SARS-CoV-2 spreads and persists throughout the organism; (ii) to compare the inflammatory infiltrates of the lung, an organ massively affected by the viral invasion, with those of the kidney, heart, and brain that are non-primary targets for the virus; (iii) to emphasize the pathological features specific for COVID-19 through comparison with non-COVID lungs and brains, and to finally investigate the role of micro-thrombosis.

However, interestingly, I could not understand how Alzheimer's disease was integrated into the subject. In my opinion, Alzheimer's disease should be completely removed from the text of this article and focused only on COVID-19 disease.

The second important issue is the liver, one of the most important immunological organs in the body, because the liver is very rich in tissue macrophages. I wonder why liver tissue was not examined in this study. When we look at the literature, one of the organs on which the most articles are written is the liver.

A few sentences about the importance of the liver should be added in the discussion section (this issue has been discussed in the articles above).

It is not appropriate to divide the discussion section into sub-headings such as A), B), C). These subheadings should be deleted. The topics to be covered can be written in paragraphs, but it is not necessary to write them in sub-headings.

Author Response

We are very grateful to the reviewer for his time, and for providing constructive and helpful advice to improve our manuscript. Here are our answers to all the questions raised by the reviewer:

REV 2

The introduction (lines 40-72) of the article should be written more fluently. So I call it the "inverted pyramid method". It is necessary to reduce the subject from the general to the specific. The article should begin with the epidemiology of the disease and then evolve towards the relevant topic. You can find the method I want to explain in the following two articles. You can benefit from the introduction of these two articles.

Sahin TT, Akbulut S, Yilmaz S. COVID-19 pandemic: Its impact on liver disease and liver transplantation. WorldJ Gastroenterol. 2020 Jun 14;26(22):2987-2999

BaÅŸkıran A, Akbulut S, Åžahin TT, Tunçer A, Kaplan K, Bayındır Y, Yılmaz S. Coronavirus Precautions: Experience of High Volume Liver Transplant Institute. Turk J Gastroenterol. 2022 Feb;33(2):145-152

In order for the readers to focus on the article, the "Study design, setting, participants and clinical data" section should be written in a more understandable and clear way.

According to your suggestions, we have made some corrections in the introduction, and method sections; please, see all the corrections in red

The authors stated that the aims of this study are (i) to describe how the organs such as lungs, kidney, heart, and brain are involved and how SARS-CoV-2 spreads and persists throughout the organism; (ii) to compare the inflammatory infiltrates of the lung, an organ massively affected by the viral invasion, with those of the kidney, heart, and brain that are non-primary targets for the virus; (iii) to emphasize the pathological features specific for COVID-19 through comparison with non-COVID lungs and brains, and to finally investigate the role of micro-thrombosis.

However, interestingly, I could not understand how Alzheimer's disease was integrated into the subject. In my opinion, Alzheimer's disease should be completely removed from the text of this article and focused only on COVID-19 disease.

Our COVID-19 case series comprises cases with neurocognitive disorders and some degree of AD pathology. In this setting, a comparison of COVID-19 cases with and without AD is very important to clarify the role of microglia, in order to determine if microglial hyperactivation is due to COVID-19 per se or is related to neurodegeneration.

The second important issue is the liver, one of the most important immunological organs in the body, because the liver is very rich in tissue macrophages. I wonder why liver tissue was not examined in this study. When we look at the literature, one of the organs on which the most articles are written is the liver.

We agree with the reviewer that the liver is one of the most important organs, but from our data as well as from the literature it does not seem particularly implicated in COVID-19. However, to clarify the reason for our choice not to consider the liver, we have included the following writing in the methodological part:

“Liver, as well as hypophysis, thyroid, spleen, adrenal glands, uterus or prostate, besides brain, lungs, heart and kidneys, were included in the routinary histopathological examination. At Hematoxylin Eosin (H&E) staining we did not observe any peculiar feature that could be related to COVID-19. Moreover, the subjects of the present study did not show any clinical sign related to a possible liver failure or impairment. Therefore, despite liver is one of the most important organs of the body, we chose to perform the study on brain, lungs, heart and kidney, for which the clinical picture and the routinary H&E staining provided the most interesting results.”

A few sentences about the importance of the liver should be added in the discussion section (this issue has been discussed in the articles above).

We add the liver to the introduction and quote the most important articles about this topic:

“Although the liver is one of the most important immunological organs in the body and alterations of liver parameters are frequently reported in COVID-19, especially in severe cases, the pathological findings are non-specific and the impairment of liver function does not appear clinically relevant in SARS-CoV-2 infection (Zhang Y 2020; Sahin TT 2020; BaÅŸkıran A 2022).”

It is not appropriate to divide the discussion section into sub-headings such as A), B), C). These subheadings should be deleted. The topics to be covered can be written in paragraphs, but it is not necessary to write them in sub-headings.

We’ll arrange the headings according to the indication of the Editorial Office.

Reviewer 3 Report

The paper of Poloni et al. demonstrated the observations regarding the involved organs along with pathological findings after SARS-CoV-2 infection. Overall strengths of the article are the illustrative materials - tables, figures, pictures. The design of the study is good, the used methods are reliable.

However, there are some issue that have to addressed to improve the manuscript:

Major - none

Minor:

1. Although it`s not specified and not investigated, it`s good to mention which SARS-CoV-2 variant(s) was(were) dominated during the period of the study in discussion. This may help established additional different features between the different virus variants.

2. Some subheadings (i.e., discussion) are just one passage in a page. Please, consider dividing into smaller paragrapghs of 5-6 lines. It`s hard to read and comprehend the presented information. Fir this, use subheadings (4.1, 4.2, 4.1.1., etc, as suggested in the template)

3. Can you make some recommendations based on your findings?

Author Response

We are very grateful to the reviewer for his time, and for providing constructive and helpful advice to improve our manuscript. Here are our answers to all the questions raised by the reviewer:

REV 3

The paper of Poloni et al. demonstrated the observations regarding the involved organs along with pathological findings after SARS-CoV-2 infection. Overall strengths of the article are the illustrative materials - tables, figures, pictures. The design of the study is good, the used methods are reliable.

However, there are some issue that have to addressed to improve the manuscript:

Major - none

Minor:

  1. Although it`s not specified and not investigated, it`s good to mention which SARS-CoV-2 variant(s) was (were) dominated during the period of the study in discussion. This may help established additional different features between the different virus variants.

We specify that all cases died of delta-variant which was dominant the period of autopsies were done.

  1. Some subheadings (i.e., discussion) are just one passage in a page. Please, consider dividing into smaller paragrapghs of 5-6 lines. It`s hard to read and comprehend the presented information. Fir this, use subheadings (4.1, 4.2, 4.1.1., etc, as suggested in the template).

We’ll arrange the headings according to the indication of the Editorial Office.

Round 2

Reviewer 1 Report

Dear Authors,

Thanks for answering the comments.

Since the pathological changes were driven by the persistent inflammation rather than the viral load, I think you may consider adding to the discussion or to the results,  whether the observed pathological changes in tissues were specific to COVID-19 complications or similar pathological changes. including the different roles of T lymphocytes and macrophages could be observed in those disorders that are majorly driven by persistent inflammation.

Thank You,

Author Response

Dear reviewer,

we better defined the specific role of T-lymphocites and macrophages-microglia in COVID-19; please, find the further changes in the discussion (red):

"The relevant presence of T lymphocytes underlines the role of these cells in the specific response where viral replication is particularly active. We can assume that such a response may be common to other pneumonia with purely viral etiology."

"Where there is no active viral replication, a non-specific macrophage-microglial response predominates, which in elderly subjects can be favored by the neurodegenerative load; hence, the importance of a comparison that included patients with AD (the most common neurodegenerative disease)."

"The pathological picture we have shown is quite different from that of viral encephalitis caused by neurotropic viruses [83], in which the presence of abundant viral antigens, abundant lymphocyte infiltrate and direct cytopathic effects are observed, as well as it occurs in the lung".

Many Thanks

Reviewer 2 Report

I still did not understand how it is associated with Alzheimer's disease. however, it is seen that the necessary changes have been made in the article.

Author Response

Dear Reviewer,

We added in the discussion a sentence to further explain the reasons for including AD pathology (red):

"Where there is no active viral replication, a non-specific macrophage-microglial response predominates, which in elderly subjects can be favored by the neurodegenerative load; hence, the importance of a comparison that included patients with AD (the most common neurodegenerative disease)."

Many Thanks